# Decline in Walking Independence and Related Factors in Hospitalization for Dialysis Initiation: A Retrospective Cohort Study

**DOI:** 10.3390/jcm11216589

**Published:** 2022-11-07

**Authors:** Yuma Hirano, Tomoyuki Fujikura, Kenichi Kono, Naro Ohashi, Tomoya Yamaguchi, Wataru Hanajima, Hideo Yasuda, Katsuya Yamauchi

**Affiliations:** 1Department of Rehabilitation Medicine, Hamamatsu University Hospital, 1-20-1 Handayama, Higashi-ku, Hamamatsu City 431-3192, Japan; 2First Department of Medicine, Hamamatsu University Hospital, 1-20-1 Handayama, Higashi-ku, Hamamatsu City 431-3192, Japan; 3Department of Physical Therapy, International University of Health and Welfare School of Health Science at Narita, 4-3, Kozunomori, Narita City 286-8686, Japan

**Keywords:** initiation of dialysis, walking independence, rehabilitation

## Abstract

Patients with chronic kidney disease require intervention planning because their physical function declines with worsening disease. Providers can work closely with patients during the induction phase of dialysis. This single-center, retrospective observational study aimed to investigate the rate of decline in walking independence during the induction phase of dialysis and the factors that influence this decline, and to provide information on prevention and treatment during this period. Of the 354 patients who were newly initiated on hemodialysis between April 2018 and January 2022, 285 were included in the analysis. The functional independence measure-walking score was used to sort patients into decreased walking independence (DWI; n = 46) and maintained walking independence (no DWI; n = 239) groups, and patient characteristics were compared. After adjusting for various factors by logistic regression analysis, we observed that age, high Charlson comorbidity index (CCI), C-reactive protein, and emergency dialysis start (EDS) were significant predictors of DWI. Even during the very short period of dialysis induction, as many as 16.1% of patients had DWI, which was associated with older age, higher CCI, higher inflammation, and EDS. Therefore, we recommend the early identification of patients with these characteristics and early rehabilitation.

## 1. Introduction

Physical function plays a role in the life outcomes of patients undergoing dialysis. Reportedly, sarcopenia status, which reflects low muscle mass, low muscle strength, or both, is associated with the survival rate in dialysis patients [1]. This implies that it is important for dialysis patients to maintain a high physical function. Moreover, the physical function of dialysis patients gradually declines as the chronic kidney disease (CKD) stage progresses from the conservative stage [2,3], and continues to decline after the introduction of dialysis. Tamura et al. evaluated the functional status of patients with seven activities of daily living (ADLs) and reported that approximately 60% of patients had a decline in functional status at three months and approximately 75% at six months after dialysis initiation [4]. Goto et al. also evaluated the functional status of 13 ADLs and instrumental ADLs (IADLs) and reported a decline in functional status in approximately 50% of patients six months after dialysis initiation [5]. In other words, the physical function of dialysis patients declined further in increments of months after dialysis introduction.

Rehabilitation from an early stage after the introduction of dialysis may reduce the decline in physical function. However, there are few reports on rehabilitation during this period. Therefore, it is necessary to first identify which patients develop functional decline. This will allow us to identify patients who require intervention at an early stage. Therefore, we focused on the induction phase of dialysis. During this phase, medical staff can work closely with patients for a certain period daily and may be able to provide rehabilitation interventions to prevent the decline in physical function. In addition, analysis of the factors contributing to the decline in physical function during this period will enable intervention from the early stages of dialysis induction. In this study, we focused on the degree of walking independence, which reflects physical function and is greatly related to ADLs, and aimed to clarify the changes in the level of walking independence during the induction phase of dialysis and the factors that influence these changes in walking independence.

## 2. Materials and Methods

This retrospective observational study was based on a single-center medical record survey. Patients newly initiated on hemodialysis at Hamamatsu University Hospital between April 2018 and January 2022 were included in this study. Exclusion criteria were as follows: patients who withdrew from hemodialysis due to improvement in renal function after hemodialysis induction, patients who did not walk independently before admission, patients who died during hospitalization, and patients who were not transferred to maintenance dialysis for other reasons. All data were extracted from the patients’ medical records. In this study, the induction phase of dialysis was defined as the period from just before admission for dialysis induction to the time of discharge. This study was approved by the Ethics Committee of Hamamatsu University School of Medicine (approval number: 22-035). Informed consent was obtained from all study participants.

Sex, age, height, weight, body mass index, medical history, blood data, length of hospital stay, presence of emergency dialysis start (EDS), and rehabilitation were recorded on admission. Blood data included creatinine, C-reactive protein (CRP), albumin, and hemoglobin levels and estimated glomerular filtration rate on admission. EDS was defined as the first dialysis session initiated within 24 h of nephrologist evaluation for life-threatening reasons, in accordance with previous studies [6,7]. Rehabilitation consisted primarily of strength training, aerobic exercise, and training in ADLs. Participants who received at least one rehabilitation session with a physical or occupational therapist during their hospitalization were counted as rehabilitation cases.

The Geriatric Nutritional Risk Index (GNRI) was used to assess nutritional impairment. The GNRI was developed as a screening measure of the risk for morbidity and mortality of disease in hospitalized older adult patients. The GNRI was calculated as (14.89 + albumin (g/dL)) + (41.7 × body weight/ideal body weight) [8], and this nutritional assessment is widely used in maintenance dialysis patients [9,10,11]. Nutritional impairment was defined as a GNRI less than 92, in accordance with a previous study [12].

Comorbidities were identified and defined using the Charlson comorbidity index (CCI) [13]. This score was developed to calculate predicted mortality from comorbidities on admission. In accordance with previous studies, CCI was divided according to higher or lower than the average [14]. High-CCI participants had a score of 4 or higher. In a previous study of dialysis patients, participants with a CCI of 4 or higher were reported to have a significantly higher mortality rate than those with a CCI of 3 or lower [15].

The gait item (functional independence measure (FIM)-walking score) of the FIM was used as an index to assess walking independence. This rating chart was developed in the U.S. as the Unified Data System’s capacity-loss rating method and reflects the patient’s “ADLs” that are being done. The FIM-walking score was evaluated on a 7-point scale ranging from independent to full assistance. This has been widely used in previous studies for functional disability assessments [16,17,18]. To ensure the reliability of the assessment, two physiotherapists (YH and WH) obtained the scores independently. Any discrepancies between them were resolved through discussion and agreement. In this study, the FIM-walking score was used to classify patients as having decreased walking independence (DWI) or maintained walking independence (no DWI). DWI was defined as “a decrease in the FIM-walking score during the hospitalization period” or “an improvement in the FIM-walking score during the hospitalization period but a decrease compared to just before admission”.

### Statistical Analysis

Descriptive statistics were used for demographic and disease-related data. Inferential statistical analysis with t-test, chi-square test, and Mann–Whitney U test was performed to examine differences in clinical disease information, patient demographics, nutritional status, comorbidities, and blood data between patients with different walking independence statuses. Univariate and multivariate logistic regression analyses were used to determine factors affecting gait independence. Items with correlation coefficients >0.80 were excluded to remove the effects of interrelated factors and multicollinearity between independent variables. Independent variables were analyzed using two models. In Model 1, we entered items of clinical significance from among variables that showed significant differences (*p* < 0.001) between the two groups, referring to previous studies [1,19,20,21]. In doing so, items that could be intermediate variables for causality (length of hospitalization, FIM-walking score at hospitalization, number of rehabilitations) were excluded. Consequently, age, CCI, CRP, and EDS were entered as independent variables. In Model 2, in addition to the variables in Model 1, sex and nutritional disorders were included as potential confounding variables [3,22]. Statistical significance was set at *p* < 0.05. IBM SPSS version 26 was used for data analysis.

## 3. Results

### 3.1. Characteristics of Participants

Of the 354 patients, 285 were included in the analysis, and 69 were excluded. After classification, 46 patients were assigned to the DWI and 239 to the no DWI (Figure 1). The mean age of the participants was 68.4 years (standard deviation (SD) = 14.4); 66.7% were men, and 33.3% were women. Of the participants, 74.0% had hypertension, 41.4% had diabetes, 33.0% had hyperlipidemia, and 43.2% had a history of cardiovascular disease. The median length of hospitalization was 14 days, and 16.1% of the patients had DWI. From admission to discharge, the percentage of FIM-walking score 1 decreased and the percentage of 7 increased (Figure 2). Of the patients, 37.2% underwent EDS. Acute exacerbation of CKD was the reason for EDS in 74.5% of cases, AKI in 17.0%, and other reasons in 8.5%. Other reasons included sepsis and multiorgan failure. In addition, approximately 8% of patients with planned dialysis start and approximately 30% of patients with EDS showed a decrease in walking independence (Figure 3). Only 29.5% of patients were rehabilitated during hospitalization. The mean GNRI was 78.2 (SD = 27.8), and 57.4% of the patients were nutritionally impaired (Table 1).

### 3.2. Differences between Patients with and without DWI

Patients with DWI were older (*p* < 0.001) and had a longer length of stay (*p* < 0.001). They also had higher rates of rehabilitation (*p* < 0.001), EDS (*p* < 0.001), and nutritional disorders (*p* < 0.05), and a high CCI (*p* < 0.001). Blood tests showed higher CRP levels (*p* < 0.001) (Table 2).

### 3.3. Factors Associated with DWI

Univariate logistic regression analysis showed that a reduction in walking independence was significantly associated with age (odds ratio (OR): 1.075, 95% confidence interval (CI): 1.040–1.112), high CCI (OR: 4.987, 95% CI: 2.305–10.788), CRP (OR: 1.296, 95% CI: 1.142–1.471), EDS (OR: 5.097, 95% CI: 2.568–10.113), and malnutrition (OR: 2.139, 95% CI: 1.047–4.370). Multivariate logistic regression analysis showed that age (OR: 1.089, 95% CI: 1.045–1.135), high CCI (OR: 4.540, 95% CI: 1.874–11.001), CRP (OR: 1.163, 95% CI: 1.049–1.291), and EDS (OR: 2.720, 95% CI: 1.217–6.080) remained significant predictors. Furthermore, after adjusting for factors in Model 2, age (OR: 1.123, 95% CI: 1.067–1.182), high CCI (OR: 4.706, 95% CI: 1.768–12.522), CRP (OR: 1.180, 95% CI: 1.057–1.317), and EDS (OR: 2.957, 95% CI: 1.239–7.059) remained significant predictors (Table 3).

## 4. Discussion

In this study, we investigated the decline in walking independence during the induction phase of dialysis and identified factors associated with this decline. The analysis showed that 16.1% of the patients had DWI during the induction phase of dialysis, which lasted approximately 14 days. Risk factors for DWI include older age, more severe comorbidities, higher inflammatory levels, and EDS. Interestingly, approximately 8% of patients with a planned dialysis start also had DWI. This is an important finding as it indicates that even patients in stable general conditions are at risk of a short-term decline in walking independence.

Patients with CKD are more likely to have reduced ADLs through loss of muscle mass, and the presence of CKD has been reported to be independently associated with functional impairment, IADL impairment, and basic ADL impairment [23]. The prevalence of sarcopenia in patients with CKD is high (approximately 50%) and is associated with a reduction in physical function [24,25,26]. In addition, patients with CKD have decreased muscle mass due to increased protein catabolism and nutrient loss due to dialysis, leading to impaired ADLs and IADLs [27]. Similarly, during the induction phase of dialysis, functional status decreases between three and six months after dialysis induction [4,5]. This suggests that the physical function and ADLs of patients with CKD may deteriorate from months to years after the introduction of dialysis.

On the other hand, this study focused on the induction phase of dialysis and, therefore, examined an even shorter period, approximately 14 days. It is clinically important to note that 16.1% of the patients had a decrease in walking independence during this short period. Early rehabilitation after dialysis induction is a potential solution to this problem. In patients with conservative CKD and those undergoing stable dialysis, exercise has been shown to improve muscle strength, exercise tolerance, and quality of life [28,29]. The effectiveness of early rehabilitation has been reported in other areas. Randomized controlled trials of patients with acute stroke have reported that early rehabilitation intervention after stroke onset was associated with a shorter time to independent ambulation and better physical function and ADLs at three and 12 months [30]. Furthermore, early rehabilitation of patients with acute nephrotic syndrome has been reported to improve their motor endurance and SF-36 physical function items without adversely affecting their disease status [31]. However, few studies have examined the effectiveness of rehabilitation during the early post-dialysis period. In this study, only approximately 30% of patients received rehabilitation. Patients immediately before dialysis induction showed a marked decrease in physical activity due to uremia, fluid overload, and renal anemia. Thus, early rehabilitation interventions to correct this decline in physical activity may be beneficial in preventing disuse syndrome and maintaining ADLs.

Our study demonstrated that older age, severity of comorbidities, higher inflammatory levels, and EDS were associated with decreased walking independence.

Generally, ADL independence in older adults decreases with age, which is also true for dialysis patients [5]. In addition to aging, CKD is known to exacerbate frailty and sarcopenia [32], and aging is a very important factor affecting ADLs in chronic inflammatory diseases.

Population aging has led to an increase in the number of patients with comorbidities, with approximately 65% of older people having three or more comorbidities [33]. The CCI is the most commonly used measure to assess comorbidity and is also commonly used in patients with conservative CKD and dialysis patients [34,35,36]. In dialysis patients, CCI is a strong predictor of mortality, and in older patients with conservative CKD, a higher comorbidity index is associated with functional decline [21,37]. Comorbidities are associated with the severity of depression, anxiety, and fatigue, which reduce the amount of physical activity in patients. This may decrease physical function, which in turn affect the degree of walking independence [38].The presence of comorbidities has received increasing attention worldwide [39] and should also be noted in patients undergoing dialysis induction.

Inflammatory markers such as interleukin-6 and tumor necrosis factor-α increase the factors associated with skeletal muscle destruction in dialysis patients, such as myostatin [40]. They are also associated with low muscle strength in patients on dialysis [1], and inflammation has been implicated in impaired physical function, corroborating previous studies. This inflammation improves with regular exercise [41] and is an important factor that may be targeted for rehabilitation therapy in the future.

Patients with CKD need to be managed so that they can be transferred to dialysis in the best possible condition. However, dialysis is often initiated on an emergency basis [20]. Furthermore, a large epidemiological study in France reported that EDS has an independent impact on mortality [42]. The main causes of EDS are acute systemic diseases, such as cardiovascular disease, sepsis, and autoimmune diseases [43]. Therefore, EDS is an important factor affecting poor prognosis, representing the poor general condition of patients, and in this study, it may have affected the level of walking independence by reflecting the underlying poor general condition of the patients.

It is clinically difficult to provide rehabilitation interventions for more than 40,000 patients undergoing dialysis induction annually in Japan. Therefore, it is necessary for healthcare providers to know in advance which characteristics of patients with CKD need rehabilitation during the induction phase of dialysis and to identify and intervene in patients with a high need for rehabilitation at an early stage. Patients with the characteristics identified in this study (i.e., older age, more comorbidities, higher inflammation, and EDS) are at a high risk of DWI in a short period of time, and it is important to start rehabilitation intervention at an early stage.

The strength of our study is that we focused on a short period during dialysis induction. Similar reports on patients with CKD prior to dialysis conversion and on stable dialysis patients are scattered. However, only few reports describing this period have been published. This study suggests that this period may have an important impact on patients’ future ADLs and calls for new interventions, including rehabilitation.

Our study had several limitations. First, some confounders remained unmeasured. Although we attempted to cover the adjustment for confounders related to physical function reported in previous studies, more detailed adjustments need to be considered in future prospective studies. Second, the FIM-walking score assessment was based on medical records, which may be inaccurate for estimating walking independence. To address this issue, two physiotherapists independently conducted the assessment, and we attempted to make it as objective as possible. Third, the GNRI may not accurately reflect the nutritional status of patients. During the induction phase of dialysis, patients may be overweight due to fluid retention, which may lead to an overestimation of the GNRI. However, an accurate nutritional assessment index has not yet been established for the induction phase of dialysis. Therefore, the GNRI, an established index for maintenance dialysis patients, was used in this study. Fourth, 19 of the 69 patients excluded died during hospitalization. The possibility that the exclusion of these 19 cases could be a confounding factor cannot be ruled out. Fifth, our study failed to consider the impact of medical restraints on DWI. Some patients are at risk for dialysis catheter removal due to delirium, insubordination to medical staff, or other reasons. Such patients are subject to medical restraints during hospitalization, which may be one of the risk factors for DWI [44]. However, the hospital has implemented initiatives to eliminate physical restraints, and only six of 285 patients (2.1%) were identified as being physically restrained. Therefore, this study could not clearly examine the impact of physical restraints on DWI.

## 5. Conclusions

Even during the very short period of dialysis induction, as many as 16.1% of patients had DWI, which was associated with older age, higher CCI, higher inflammation, and EDS. Therefore, we recommend the early identification of patients with these characteristics and early rehabilitation to reduce the decline in physical function and ADLs.

## Figures and Tables

**Figure 1 jcm-11-06589-f001:**
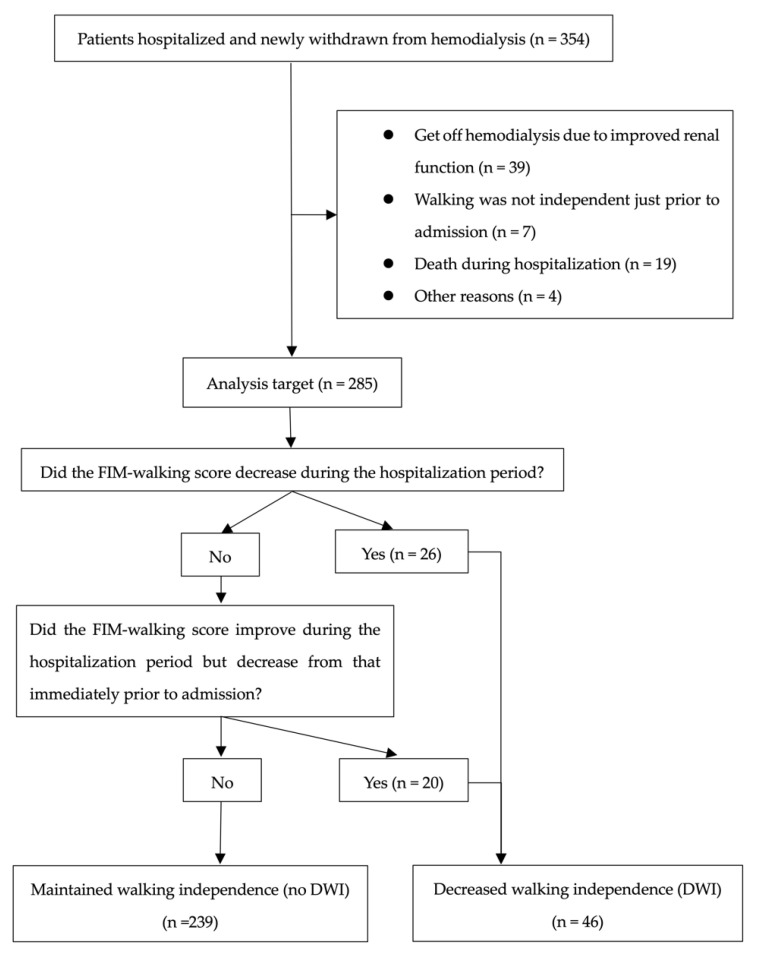
Selection of participants for analysis and assignment to the two groups.

**Figure 2 jcm-11-06589-f002:**
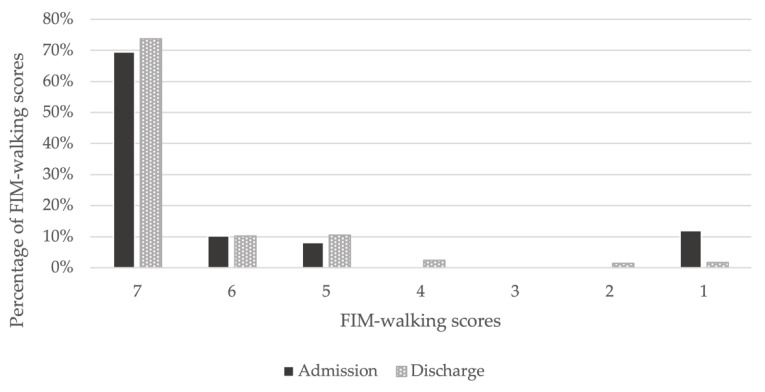
Changes in FIM-walking scores between admission and discharge. On admission, 69% had an FIM-walking score of 7, which increased to 74% at discharge. An FIM-walking score of 1 was present in 12% of patients on admission, but decreased to 2% at discharge.

**Figure 3 jcm-11-06589-f003:**
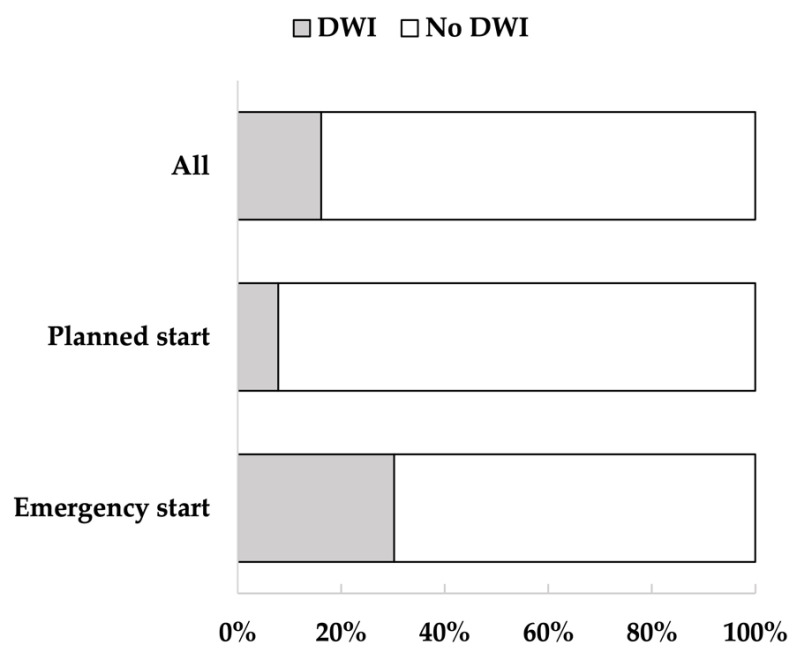
Percentage of decreased walking independence (DWI) in all patients, patients with planned dialysis start, and patients with emergency dialysis start (EDS). The figure shows the percentage of patients with DWI. Walking independence decreased in 16% of all patients, 8% of patients with planned dialysis start, and 30% of patients with EDS.

**Table 1 jcm-11-06589-t001:** Patients’ characteristics (N = 285).

Variable	N, Mean, Median
Sex	
Male [N, (%)]	190 (66.7)
Female [N, (%)]	95 (33.3)
Age (years)	68.4 ± 14.4
Median length of hospitalization (days)	14 (12–28)
DWI [N, (%)]	46 (16.1)
Hypertension [N, (%)]	211 (74.0)
Diabetes mellitus [N, (%)]	118 (41.4)
Hyperlipidemia [N, (%)]	94 (33.0)
Previous history of CVD [N, (%)]	123 (43.2)
Number of rehabilitations [N, (%)]	84 (29.5)
Emergency dialysis start [N, (%)]	106 (37.2)
Creatinine (mg/dL)	7.7 ± 3.0
eGFR (mL/min/1.73 m^2^)	7.1 ± 6.3
Geriatric Nutritional Risk Index	78.2 ± 27.8
Malnutrition	159 (57.4)

Data are expressed as percentage in parentheses or as the mean ± SD or median (IQR). *CVD*, cardiovascular disease; *eGFR*, estimated glomerular filtration rate. Nutritional impairment was defined as a GNRI less than 92.

**Table 2 jcm-11-06589-t002:** Differences between DWI and no DWI patients.

	DWI (N = 46)	No DWI (N = 239)	*p*-Value
Male [N, (%)]	35 (76.1)	155 (64.9)	0.139
Age (years)	78.5 ± 8.2	66.4 ± 14.6	<0.001 *
Height (cm)	158.0 ± 8.1	159.6 ± 9.4	0.287
Body weight (kg)	56.2 ± 10.0	61.2 ± 15.9	0.127
BMI (kg/m^2^)	22.5 ± 3.6	23.9 ± 5.1	0.213
Median length of hospitalization (days)	34 (26–64)	13 (12–22)	<0.001 *
FIM-walking score at hospitalization < 6 [N, (%)]	27 (58.7)	31 (13.0)	<0.001 *
Number of rehabilitations [N, (%)]	33 (71.7)	51 (21.3)	<0.001 *
EDS [N, (%)]	32 (69.6)	74 (31.0)	<0.001 *
Malnutrition [N, (%)]	31 (72.1)	128 (54.7)	0.034 *
High CCI [N, (%)]	37 (80.4)	108 (45.2)	<0.001*
CRP (mg/dL)	0.43 (0.16–0.57)	0.08 (0.03–0.17)	<0.001 *
Alb (g/dL)	3.5 (3.1–3.8)	3.5 (3.2–3.9)	0.145
Hb (g/dL)	9.4 (9.1–10.0)	9.6 (9.0–10.3)	0.645

Data are expressed as percentage in parentheses or as the mean ± SD or median (IQR). *BMI*, body mass index; *EDS*, emergency dialysis start; *CCI*, Charlson comorbidity index; *CRP*, C-reactive protein; *Alb*, albumin; *Hb,* hemoglobin. * Statistically significant.

**Table 3 jcm-11-06589-t003:** Factors related to decreased walking independence.

	Univariate Regression			Multiple Logistic					
				Model 1			Model 2		
	OR	95% CI	*p*-Value	OR	95% CI	*p*-Value	OR	95% CI	*p*-Value
Age	1.075	1.040–1.112	<0.001 *	1.089	1.045–1.135	<0.001 *	1.123	1.067–1.182	<0.001 *
High CCI	4.987	2.305–10.788	<0.001 *	4.540	1.874–11.001	0.001 *	4.706	1.768–12.522	0.002 *
CRP	1.296	1.142–1.471	<0.001 *	1.163	1.049–1.291	0.004 *	1.180	1.057–1.317	0.003 *
EDS	5.097	2.568–10.113	<0.001 *	2.720	1.217–6.080	0.015 *	2.957	1.239–7.059	0.015 *
Sex (male)	0.580	0.280–1.201	0.142				0.839	0.336–2.093	0.706
Malnutrition	2.139	1.047–4.370	0.037 *				1.535	0.645–3.654	0.332

Multivariate logistic regression analysis. *CCI*, Charlson comorbidity index; *CRP*, C-reactive protein; *EDS*, emergency dialysis start; *OR*, odds ratio; *CI*, confidence interval. Model 1: Variables with significant differences (*p* < 0.001) that were clearly different between the two groups were selected. However, variables (length of hospitalization, FIM-walking score at hospitalization, number of rehabilitations) that could be “intermediate variables in a causal chain of causes and effects” were excluded. Model 2: Sex and malnutrition were used as adjustment variables. * Statistically significant.

## Data Availability

The data presented in this study are available on request from the corresponding author. The data are not publicly available because they are the property of the Institute of Hamamatsu University Hospital, Japan.

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
