# Peer review of "Decline in Walking Independence and Related Factors in Hospitalization for Dialysis Initiation: A Retrospective Cohort Study"

_jcm, 2022, doi:10.3390/jcm11216589_

Round 1

Reviewer 1 Report

Dear the authors,

The authors evaluated the rate of decline in walking independence in 285 incident hemodialysis patients during their hospitalization. As a result, forty-six patients (16.1%) showed decrease of walking independence (DWI). Significant predictors of DWI were age, high-CCI, CRP, and EDS. As authors recommended and concluded, I agree it is very important to identify patients with high risks of DWI. Authors evaluated some factors. However, from the point of clinical view, the assessment authors provided seems to be somewhat weak. I think patients at dialysis induction phase have several physical problems other than authors evaluated. 

1.     Please provide more precise data about functional independence measure (FIM)-walking score in patients with DWI (FIM walking scores at admission and at discharge as dot plot figure) for better understanding for readers. FIM-walking score 7 only indicates walking independence. However, readers might want to know the real scores, 1,2,3,4,5,6 at admission and at discharge (dynamic change of FIM walking score).

2.     Patients with emergency dialysis start (EDS) might not have A-V fistula beforehand.

Did all patents with EDS not have A-V fistula and did they need to undergo dialysis catheter insertion?

3.     Please provide the reason of EDS. 

4.     Patients with EDS might have oxygen demand due to uremic lung or congestive heart failure. Please analyze the association of these factors with DWI.

5.     I also suppose that some patients might have delirium or disobey medical staff, or have risk for dialysis catheter removal. They might be treated with medical restraint during their hospitalization. I suppose medical restraint might be the most important associating factor for DWI. Please provide the rate of restraint during hospitalization in DWI patients and no-DWI patients, and please analyze again with medical restraint as one of the risk factors for DWI.

6.     Did authors not evaluate Berthal Index (BI)? If possible, please provide the BI in the subjects at the administration and at discharge.

Reviewer 2 Report

In the manuscript by Hirano et al the authors investigate the association of decline in walking independence during dialysis induction and related factors. 285 patients were included, and a walking score was used to evaluate the walking independence. 46 patients showed a reduction in walking independence. The authors found that age, CCI, CRP and EDS were significant predictors of walking independence. This is an interesting and well-written study. I have some comments which may improve the manuscript:

23/24: During the dialysis induction phase (lasting approximately 14 days), 16.1% of the patients showed a decrease in walking independence. à This is a redundancy of the sentence before (46 of 285) and can be deleted or summarized.

26: CCI: should be explained at first use

27: EDS: should be explained at first use

28: The conclusion could be stronger and more specific. Compare conclusion at the end of the manuscript.

Abstract: The study design (single-center retrospective observational study) could be included in the abstract.

64: Patients who died during follow-up were excluded from analysis: could this be a confounding factor?

87: How is High/Low-CCI defined? Which average, of hemodialysis patients?

127-129: check sentence

129: How is rehabilitation defined?

142: The formatting of the table could be improved (many empty fields). All information (N, mean, median) could be included in one column. E.g. Male [N, (%)] – 190 (66.7); Age – 68.4 ± 14.4; Length of hospitalization – 14 (12 -28).

147: Definition of Malnutrition would be helpful in the table description to directly understand the information.

158: The log values are not very helpful to understand the blood parameters. I suggest including the median values if not normally distributed.

172: Which and how were the factors selected for adjustment?

181-183: This is not clear. Which factors were excluded?

235-237: This sentence is not very clear in this context (very unspecific).

238-239: Which factors increase?

240: Reportedly – is there evidence or not?

295: Why is data availability statement not applicable?

Round 2

Reviewer 1 Report

Dear the authors,

You appropriately replied reviewer's comments, and additional data analysis was done.  

Changes of FIM-walking scores at administration and discharge is very suggestive. Patients with score 7 increased and score 1 decreased at discharge (suggestive of rehabilitation success).   

Reviewer also pointed that medical restraint might be a major important factor for DWI. However, your hospital has strong policy to eliminate physical restraint. Therefore, only 2.1% of the patients were found to be physically restrained. It is very good thing, but this is not available in all hospital. Authors might comment about this issue in discussion. 

Author Response

Response : Thank you for your valuable comments. We have taken your point and addressed this concern by adding these sentences in the “Discussion” section (page 9, lines 291–297):

Fifth, our study failed to consider the impact of medical restraints on DWI. Some patients are at risk for dialysis catheter removal due to delirium, insubordination to medical staff, or other reasons. Such patients are subject to medical restraints during hospitalization, which may be one of the risk factors for DWI [44]. However, the hospital has implemented initiatives to eliminate physical restraints, and only 6 of 285 patients (2.1%) were identified as being physically restrained. Therefore, this study could not clearly examine the impact of physical restraints on DWI.
